# Multifunctional Edible Amaranths: A Review of Nutritional Benefits, Anti-Nutritional Factors, and Potential in Sustainable Food Systems

**DOI:** 10.3390/foods15010130

**Published:** 2026-01-01

**Authors:** Svetoslava Terzieva, Stanka Baycheva, Milena Tzanova, Teodora Ivanova, Dessislava Dimitrova, Neli Hristova Grozeva

**Affiliations:** 1Department of Biological Sciences, Faculty of Agriculture, Trakia University, Students’ Campus, 6000 Stara Zagora, Bulgaria; milena.tsanova@trakia-uni.bg (M.T.); n.grozeva@trakia-uni.bg (N.H.G.); 2Department of Food Technology, Faculty of Technics and technologies, Trakia University, 38 Graf Ignatiev Str., 8602 Yambol, Bulgaria; stanka.baycheva@trakia-uni.bg; 3Department of Plant and Fungal Diversity and Resources, Institute of Biodiversity and Ecosystem Research, Bulgarian Academy of Sciences, 1113 Sofia, Bulgaria; tai@bio.bas.bg (T.I.); dessidim3010@gmail.com (D.D.)

**Keywords:** *Amaranthus* spp., nutritional value, anti-nutritional factors, healthy food

## Abstract

In recent decades, species within the genus *Amaranthus* L. (amaranth) have garnered growing global interest due to their exceptional nutritional value, functional properties, and agricultural versatility. Traditionally consumed as leafy vegetables or pseudo-cereals, several *Amaranthus* species are now receiving renewed attention in the context of the development of modern functional foods. This review evaluates the data on nutritional composition, health-promoting properties, and potential applications of *Amaranthus* spp. in sustainable food systems in peer-reviewed publications from the last 25 years. Amaranth is rich in high-quality proteins, essential amino acids, dietary fibre, vitamins, and minerals, positioning it as a significant factor in addressing malnutrition and enhancing food security. Furthermore, its bioactive compounds, such as flavonoids, phenolic acids, and peptides, exhibit antioxidant, anti-inflammatory, and hypocholesterolemic activities, suggesting its potential as a part of healthy diets, alleviating the risk of non-communicable diseases. The presence of anti-nutritional factors, including saponins, phytates, and oxalates, has also been explored, with implications for nutrient bioavailability and overall health effects. In addition to its nutritional advantages, *Amaranthus* spp. demonstrate strong adaptability to diverse climatic conditions, thus performing as a crop resilient under climate stress. Their olfactory and sensory attributes are also considered important for consumers’ acceptance and market integration. By synthesising traditional knowledge and contemporary scientific research, this review underscores the potential of *Amaranthus* spp. as a multifunctional food source that could support health promotion, climate resilience, and agricultural sustainability.

## 1. Introduction

The quality and diversity of the consumed food have a strong impact on human health. In light of the global population growth and increasing nutritional demands, there is an increasing need for food systems that not only ensure food security but also provide access to nutrient-dense, protein-rich foods. In recent decades, culinary diversity has gained importance due to its relevance to both nutrition and cultural heritage, as well as its role in fostering dietary adaptation to changing environmental and social conditions.

The wide array of food products, ingredients, recipes, and preparation techniques across different cultures and regions reflects not only reflects traditional knowledge but also serves as an integral component of human health and well-being. Simultaneously, climate change and extreme weather events pose serious threats to agricultural sustainability and global food security—challenges that affect developing and developed countries alike. In this context, greater attention must be given to climate-resilient crops that offer nutritional benefits and possess culinary versatility [1].

Consumer interest in functional foods, those with proven or potential health benefits beyond basic nutrition, has surged in recent years. These foods are recognised for promoting overall health and reducing the risk of chronic, non-communicable diseases. Numerous plant-based materials, including stems, fruits, leaves, and barks, as well as traditional herbal preparations, are widely utilised in ethnomedicine and preventive healthcare worldwide [2]. Scientific interest in plant-based functional ingredients, particularly gluten-free pseudo-cereals, has intensified due to their nutritional value, bioactive compound content, and potential for addressing dietary restrictions and deficiencies [3].

Among these crops, species of the genus *Amaranthus* L. (amaranths) stand out for their high nutritional value, exceptional adaptability to diverse agro-climatic conditions, and longstanding culinary traditions. Amaranth have often been classified as a “superfood”, a term reserved for natural foods rich in essential nutrients, low in calories, and associated with positive health outcomes [4]. The genus *Amaranthus*, belongs to the family Amaranthaceae and comprises over 70 species that are widely spread across tropical, subtropical, and temperate regions [5]. Representatives of the genus are annual or perennial herbaceous plants, which may be glabrous or covered with glandular or multicellular hairs [6,7]. While many species are considered weedy or ruderal, several have been cultivated for centuries as food, fodder, or ornamental plants by civilisations such as the ancient Greeks and the Incas [8].

Despite their nutritional, functional, and agroecological potential, amaranths remain underutilised in mainstream industrial food products. Researchers have emphasised the need to develop appropriate technologies for their incorporation into modern food systems, particularly as a raw material in the production of functional foods. Such efforts could contribute significantly to addressing challenges related to malnutrition, climate resilience, and global food security [9,10]. The integration of amaranths into contemporary food formulations may also broaden the scope and availability of functional products on the market [11].

This review summarises current knowledge on nutritional composition, bioactive compounds, health benefits, and technological potential of the most widely distributed *Amaranthus* spp. worldwide. An uneven scientific interest in the species included in the review was registered. Therefore, the goal of this study is to highlight the priorities for future research related to the position of amaranths as a functional food and resilient crop in changing climate conditions.

## 2. Materials and Methods

This review was conducted following a systematic approach to literature identification and synthesis, adhering to the PRISMA 2020 (Preferred Reporting Items for Systematic Reviews and Meta-Analyses) guidelines. The search spanned a 25-year period (2000–2025) to capture the evolution of *Amaranthus* research—from fundamental nutritional characterisation to advanced biotechnological and sustainability applications.

The search was executed across four primary databases: Web of Science, Scopus, PubMed, and Google Scholar. To maximise the retrieval of relevant studies, the strategy employed Boolean operators to combine the terms “*Amaranthus*” or “amaranth” with specific thematic clusters:
Nutrition: “nutritional value,” “proximate composition,” and “amino acids.”Bioactivity: “bioactive compounds,” “phenolics,” and “antioxidants.”Safety: “anti-nutritional factors,” “oxalates,” and “phytates.”Sustainability: “sustainable agriculture,” “climate resilience,” and “food security.”

To complement the database search, a “snowballing” technique was applied. This involved a manual review of the reference lists from seminal papers and high-impact reviews to ensure no critical evidence was overlooked.

The selection process was governed by stringent eligibility criteria designed to ensure the quality and relevance of the evidence:Inclusion Criteria: Peer-reviewed original research articles and book chapters published in English were included. Eligible studies were required to provide quantitative data on the nutritional, bioactive, or agronomic profiles of edible *Amaranthus* species, with a clear focus on food science or sustainable production.Exclusion Criteria: Non-peer-reviewed materials—such as conference abstracts, theses, and technical reports—were excluded. Additionally, studies focusing exclusively on ornamental varieties or non-edible weed species were omitted to maintain the review’s focus on human nutrition and agriculture.

Following the removal of duplicate records, the remaining studies underwent a structured, multi-stage screening process. Titles and abstracts were initially assessed for relevance, followed by full-text evaluation. In total, 224 articles were assessed at the full-text level, of which 191 reports were successfully retrieved and evaluated for methodological rigour and compliance with the eligibility criteria. Of the final 191 included studies, approximately 40 contributed quantitative data on nutritional composition, around 22 focused on experimentally validated biological activities, and approximately 25 addressed ethnomedicinal uses. These thematic groups partially overlap and therefore represent analytical subsets of the total body of evidence.

The systematic review was conducted in accordance with the PRISMA 2020 guidelines (Appendix A), and the completed PRISMA 2020 checklist is provided in the Appendix A.

## 3. Results and Discussion

*Amaranthus* spp. serve a dual function: they are cultivated as pseudocereals in different parts of the world, primarily in the Americas, and as leafy vegetables that are traditionally consumed across Africa, Asia, and Europe [11]. The seeds can be eaten directly or processed into a variety of food products, while the nutrient-rich leaves are often incorporated in traditional dishes [12,13]. Our literature search has highlighted 15 *Amaranthus* species that have been in the focus of the research defined topics over the last 25 years (Figure 1). *Amaranthus cruentus* L. has attracted the highest research interest, which has increased in the last five years. The interest in the nutritional and pharmacological value of wild, including weedy, species has attracted less attention as those species are rarely used as food and animal feed by local communities, as well as in some traditional medical systems [14]. Most *Amaranthus* species have edible leaves, with *A. blitum* L. (syn. *A. lividus* L.), *A. viridis* L. (syn. *A. gracilis* Desf.), and *A. tricolor* L. (syn. *A. gangeticus* L.) among the most popular and widely consumed leafy amaranths in Africa and Asia. However, they are researched less often in comparison to those cultivated for their grain (like *A. cruentus* and *A. hypochondriacus* L., the latter ranking third in Figure 1). This tendency could be related to the intensification of the interest in climate-smart grains along with the growing demand for pseudocereals in recent years [12,15]. Regrettably, in many cases, researchers overlook the botanical identity of crops (found in the literature only as *Amaranthus* spp.) that are offered in bulk or as already processed products, like flours, both to consumers and industry. Thus, valuable data that are species-referenced are lost, i.e., variety of traits, especially those related to ecological and climatic conditions that might have an impact on the chemical composition of the biologically active compounds in the plants, are overlooked. Such an approach might be explained by the challenges in the taxonomic identification of *Amaranthus* species, which is due to their phenotypic plasticity and high hybridisation potential, but also to an overall topical fragmentation of current research.

### 3.1. Amaranthus spp. as a Crop and Wild Edible Plant—Alimental Insights and Cultural Perspectives

*Amaranthus cruentus*, *A. hypochondriacus*, and *A. caudatus* L. are primarily grown for their seeds in tropical and subtropical regions [16,17,18]. The leaves and whole herbage of young plants of most amaranths are edible, but simultaneous production of both grain and leaves was advised only for small-scale farming, as defoliation reduces grain yields (Figure 2, Table 1) [19]. *Amaranthus* are characterised by a short growing season which makes it suitable for multipurpose utilisation. Their high ecological adaptability also allows them to thrive well both under field conditions and in more challenging environments [20,21,22]. Grain and leafy amaranths display diverse morphological and physiological traits, which might be associated with their remarkable capacity to adapt to varying environmental conditions [23]. Species such as *A. retroflexus* L. and *A. tricolor* are particularly tolerant to drought and heat, making them well-suited for cultivation in water-limited and marginal soils [24,25,26]. Consequently, *Amaranthus* spp. are increasingly recognised as promising candidates for sustainable agriculture in the face of climate change. On the other hand, these promising traits of leafy amaranths are one of the reasons for their underrepresentation in crop systems as many believe species like *A. thunbergii* Moq., *A. grecizans* L., *A. spinosus* L., *A. deflexus* L., *A. viridus* L., *A. hybridus* L., etc., could be easily gathered from the wild [27]. However, in the frame of heavy environmental pollution caused by intensive agriculture and transport, this option could be quite risky, especially in urban and peri-urban regions.

Being widely utilised across the world, though primarily within small communities, amaranths have been cultivated since ancient times and continue to be valued in the modern era [28,29,30,31,32,33,34]. Species of *Amaranthus* have played a prominent role in the cultural and dietary practices of various ancient civilisations in Mesoamerica, the Andean region, parts of Asia, and the Mediterranean basin [35]. Archaeobotanical and historical evidence suggests that amaranths were domesticated over 5000 years ago and served not only as a staple food crop, but also as a sacred plant integrated into religious and ritualistic traditions [36]. In pre-Columbian societies of the Aztecs and the Maya, grain amaranths (*A. cruentus* and *A. hypochondriacus*) were extensively cultivated and highly valued both for their nutritional richness and symbolic role in religious ceremonies. The seeds were incorporated into traditional foods and beverages and, in some ritual contexts, mixed with honey or even human blood to form symbolic offerings known as *zoale*. These offerings played a central role in religious ceremonies dedicated to deities such as Huitzilopochtli, the god of war and the sun [8]. Consequent to the Spanish conquest in the 16th century, amaranth cultivation declined significantly in order to abolish indigenous religious practices; thus, the cultural continuity was broken, and the crop was abandoned [37]. However, the use of grain amaranths as food has survived in certain local communities, e.g., amaranth seeds are used in traditional sweets and desserts, such as *alegría* in Mexico [38]. In India, the seeds known locally as *rajgira* or *ramdana* are commonly consumed during religious festivals, and are often processed into sweets or savoury snacks [39].

**Table 1 foods-15-00130-t001:** *Amaranthus* species as food: edible plant parts and main culinary uses.

Species	Edible Plant Parts	Main Food Uses	References
*A**. albus* L.	aerial parts	vegetable	[40]
*A. blitum* L. (syn. *A. lividus* L., *A. viridis* All.)	aerial parts, leaves	vegetable, cooked leaf vegetable (fresh and dried)	[41,42,43,44,45,46]
*A. caudatus* L. (syn. *A. mantegazzianus* Passer)	leaves, seeds	biscuits, bread, crackers, flour, pasta, popping of amaranth grain, saag, vegetable	[47,48,49,50,51]
*A. cruentus* L.	leaves, seeds, stems	canned amaranth leaves, dried condiments, expanded seeds (popping), flakes, flour, fresh salads and dishes, instant noodles	[10,44,51,52,53,54,55,56,57,58]
*A. dubius* Mart. ex Thell.	leaves	steamed bread	[44,59,60]
*A. graecizans* L.	leaves	vegetable cooked leaf	[61]
*A. hypochondriacus* L.	amaranth sprouts, leaves, seeds	amaranth flour, amaranth seeds, baked products, bread, breakfast items, candies, chapati popped, fresh salads, high-antioxidant capacity beverages, molasses, pasta, roasted amaranth flour (beverage preparation), soups, vegetable cooked leaf	[44,61,62,63,64,65,66,67,68]
*A. paniculatus* L.	aerial parts	vegetable	[41,64]
*A. retroflexus L.*	aerial parts, leaves, seeds, whole plant	bread, dried, flours, fresh, raw or toasted, in pie fillings	[40,69,70]
*A. spinosus* L.	leaves, roots, seeds, whole plant	flour (bread), sauces, soups, vegetables	[9,42,44,63,68]
*A. thunbergii Moq.*	leaves	fresh or dried	[44,71]
*A. tricolor* L. (syn. *A. gangeticus* L., *A. polygamus* L.)	aerial parts, leaves, seeds	biscuits, vegetables	[41,42,44,72]
*A. viridis* L.	leaves, shoots, young plants	cooked vegetable, cooked vegetables, fried vegetable, steamed vegetable	[44,73]
*Amaranthus* spp.	aerial parts, leaves, seeds, seeds and leaves	bakery, biscuits, bread (popping, steaming), cakes, candies, cassava breads, condiment, cookies, crackers, high protein beverage, maize breads, noodles formation, pancakes, pasta, puree, salads, snack bar, snack cake, soups, starch flour bakery, thickener in sauces, tortillas, vegetable, wheat flour blended with high amaranth protein content to enhance the nutritional worth of final food products such as noodles, cookies, potatoes, and breads	[17,39,62,74,75,76,77,78,79,80,81,82,83,84,85,86,87,88]

The abandonment of the grain amaranths after the Spanish conquest, could explain why, in different places in Africa, Asia and Europe, amaranths have appeared not as curious neophytes that later would become established crops like tomatoes, peppers, potatoes, and beans. Most often, the amaranths have reached the Old World as weeds travelling alongside other crops imported from the New World. In African, Asian, and European cuisines, amaranth aerial parts are consumed raw, sautéed, steamed, or incorporated into pie fillings, soups and stews. While some cultures have adopted them as prized leafy vegetables, in other places (Southeastern Europe), amaranths have remained “second-choice” wild edibles that reached tables mainly in times of war or hunger [70,89]. Aerial parts of *A. retroflexus*, *A. blitum*, and *A. viridis* are added to pie crusts or consumed as salads in the Balkans and the circum-Mediterranean area [45,70]. *Amaranthus spinosus* is traditionally consumed in Tanzania, reflecting both the cultural and nutritional significance of the genus in African diets [9]. Other species, such as *A. dubius* and *A. paniculatus*, are used primarily as leafy vegetables, while *A. albus* and *A. retroflexus* have regional culinary importance in some Asian and African communities [10,40,41]. Still, greater food variety and improved food affordability in urbanised settings were reported to contribute to the underutilisation of some of the species, like *A. spinosus*, *A. dubious*, etc., in Kenya [90].

Interestingly, nowadays amaranths have been recognised as a modern functional food due to their health-promoting chemical composition [91,92]. Chefs and health-food enthusiasts are experimenting with incorporating amaranth seeds into salads, soups, breads, and pasta. High-energy snack bars, cereals (e.g., muesli), plant protein powders, and gluten-free flours are being developed from amaranths as a response to the growing demand for gluten-free, plant-based, high-protein products. Grain amaranths have found their industrial applications in functional bakery products like pasta, pies, crackers, biscuits, pie crusts, breads and snacks formulations thanks to the protein, fibre, and mineral content of the amaranth flour [15,39,74,75,76,77,78,79,93,94,95,96,97,98]. The aerial parts and leaves of *Amaranthus* spp. are also increasingly utilised in high-protein beverages (e.g., smoothies) and functional food formulations [62,79]. The main limitations to the incorporation of higher amounts of amaranth powder are its sensorial characteristics—green colouration, strong aroma and taste that require additional technological approaches to improve consumers’ attitude [99,100].

### 3.2. Nutritional Value and Health Benefits

Amaranths are pseudocereals that are often referred to as a “superfood” due to their high content of proteins (ranging between 12% and 18%), lipids (notably omega-3 and omega-6 fatty acids), carbohydrates, dietary fibre, tocopherols, phenolic compounds, vitamins (A, C, and folate), and essential minerals such as Ca, Fe, and Mg [4,9,101,102,103,104,105,106]. Compared to other grains like wheat and rice, amaranth seeds contain significantly higher amounts of lysine, an essential amino acid often lacking in plant-based diets [93,107]. Additionally, amaranths exhibit higher antioxidant activity than rice and buckwheat, for instance [108]. This is supported by the high squalene content in some amaranth species (8.05–11.19%), which surpasses levels found in quinoa and rice bran (3.39% and 3.10%, respectively), contributing to its antioxidant capacity [109]. As a gluten-free pseudocereal, amaranth seeds are suitable for individuals with gluten intolerance or celiac disease [75,106]. Additionally, sprouts of *A. hypochondriacus* are rich in fibre and protein and demonstrate antihypertensive and antioxidant activity, reinforcing their potential as nutrient-dense functional ingredients [65]. Among the grain amaranths, *A. caudatus* exhibits strong antioxidant activity and, like other species in the genus, produces gluten-free seeds with a balanced amino acid profile and superior starch digestibility compared to traditional cereals [110]. Its seeds are rich in bioactive compounds with antidiabetic, antihyperlipidemic, and antimicrobial effects, highlighting their relevance in functional food development and chronic disease prevention [62]. Similarly, protein concentrate from *A. cruentus* contains higher levels of protein and dietary fibre as well as beneficial unsaturated fatty acids and squalene compared to seed flour. The concentrate exhibits an optimal amino acid composition and high biological value, while saponins, phytates, and trypsin inhibitors may support lipid metabolism and cardiovascular health [57]. Processing techniques such as cooking, popping, germination, and milling further enhance the mineral, vitamin, and sugar content of *A. caudatus* and *A. cruentus* seeds, thereby improving the overall nutritional and sensory quality of derived food products [51].

Recent comparative analyses have highlighted a significant nutritional and phytochemical variability among leafy *Amaranthus* species. Studies on *A. lividus* (syn. *A. blitum*) indicate that it is a rich source of proteins and bioactive compounds with pronounced antioxidant and antimicrobial activity, supporting its traditional medicinal uses [111]. Recent evaluations of leafy *Amaranthus* species reveal notable variation in their nutritional and phytochemical profiles. According to Odhav et al. [112], *A. spinosus* is a nutritionally valuable species enriched in proteins, dietary fibre, and essential minerals, supporting its role as a dense source of nutrients in traditional diets. Studies by Sarker and Oba [26] further indicate that selected red and green morphs of *Amaranthus*—including *A. tricolor*—possess elevated levels of antioxidant constituents such as chlorophylls, carotenoids, vitamin C, total phenolics, and flavonoids, which contribute to strong DPPH and ABTS^+^ radical scavenging activity. In addition, *A. tricolor* has been shown to accumulate diverse bioactive compounds, including quercetin, rutin, and other flavonoids, as well as phenolic constituents with reported hepatoprotective potential [113]. Collectively, these findings demonstrate that different *Amaranthus* species exhibit distinct patterns of nutrient and phytochemical accumulation, which underpin their high antioxidant capacity and functional value.

Accumulated literature data indicate the established bioactive properties of the investigated *Amaranthus* species, including antioxidant, anti-inflammatory, antimicrobial, and hepatoprotective activities, primarily associated with the presence of phenolic compounds, flavonoids, betalains, saponins, sterols, peptides and others (Table 2, Figure 3).

**Table 2 foods-15-00130-t002:** Experimentally Validated Bioactivities and Ethnomedicinal Uses of *Amaranthus* spp.

Species	Plant Part	Experimentally Validated Biological Activity	Ethnomedicinal Uses	References
***A. albus* L.**	aerial parts	pancreatic lipase inhibitory activity (anti-obesity potential) [114]	as food or tea, reported to influence digestion [40]	[40,114]
***A. blitum* L. (syn. *A. lividus* L., *A. viridis* All.)**	aerial parts, aerial parts (ethanol extract and fractions), leaf decoction, leaf poultice, leaves, seeds, whole plant	antioxidant and free radical-scavenging activities [41,43]; antioxidant and neuroprotective activities [115]; neuroprotective and anti-inflammatory activities [116]; antihyperglycemic and hypolipidemic activities [117]; antidiabetic and anticholesterolemic activities [118]; strong antioxidant and DNA-protective activities [119]; anthelmintic, anti-platelet and anti-coagulant activities [120]; antioxidant and anticancer activities [60]; neuroprotective, anticancer, and antibacterial effects [121].	leaves used for strangury, gonorrhoea, urinary stone removal, and as a skin tonic [42]; blood-purifier, diuretic; used in piles, strangury, dropsy, pulmonary issues, scrofula, ulcers, diarrhoea, oral/throat inflammation [122]	[41,42,43,44,60,115,116,117,118,119,120,121,122]
***A. caudatus* L. (syn. *A. mantegazzianus* Pass.)**	aerial parts, leaves, roots, seeds, whole plant	antioxidant activity [18,110]; antidiabetic and anti-cholesterolemic activities [118]; strong iron-chelating antioxidant activity [123]; antitumor effect [124]; antimicrobial activity and growth-stage-dependent toxicity [125]; strong antidiabetic, insulin-boosting activity [126]; antioxidant, antihypertensive and antidiabetic activity [127]		[18,110,118,123,124,125,126,127]
***A. cruentus* L.**	aerial parts (leaves and stems), leaves, root decoction, seeds	antioxidant potential [123]; antioxidant and anti-inflammatory activities [128]; antifungal activity [129,130]; antioxidant and xanthine oxidase inhibitory activities [131]	for anaemia [44,132]	[44,123,128,129,130,131,132]
***A. deflexus* L.**	leaves	antioxidant activity [133,134]	No specific data found.	[133,134]
***A. dubius* Mart.**	fruits, leaves	antiviral activity [121,135]; antihypertensive activity [136]	for anaemia and general health [44]; anaemia [137]; for kidney issues, anaemia, fever, bleeding, stomach ailments, and hypertension [136]	[44,121,135,136,137]
***A. hybridus* L.**	aerial parts (leaves and stems), leaves, seed extract, seeds, whole plant	antioxidant and anticancer activities [60]; strong antioxidant, anticancer, and antimicrobial properties [121]; antioxidant capacity [123]; antifungal effects [129]; strong antioxidant activity [131]; antihypertensive, antioxidant, antimicrobial, hepatoprotective, and anticancer activities [136]; antioxidant and antimicrobial potentials [138]; antioxidant, anticancer, and antimicrobial activities [60]; antioxidant activity [139]	for digestive discomfort and general weakness [44]; used to support general health and nutrition [136]	[44,60,60,121,123,129,131,136,138,139]
***A. hypochondriacus* L.**	flowers, leaves, seeds	antioxidant activity [61,123]; anticarcinogenic and antihypertensive activities [66]; anticancer and antihypertensive activities [121]; antioxidant activity [123]; antidiabetic [140]; antithrombotic, antihypertensive, antioxidant and anti-inflammatory effects [141]; anti-inflammatory and antioxidant activities [142]; antioxidant activity [143]	for digestion and general health [44]; for ulcers, diarrhoea, and oral/throat inflammation [122]	[44,61,66,121,122,123,140,141,142,143]
***A. paniculatus* L.**	aerial parts, leaves	antioxidant and free radical-scavenging activities [41]; antioxidant activity [144]	for antioxidant health benefits [144]	[41,144]
***A. retroflexus* L.**	aerial parts, inflorescences, leaves, roots	antioxidant activity [69]; antifungal and antimicrobial activities [121]; antifungal activity [129]; antioxidant activity [123,145]; cytotoxic activity and cytotoxic effect on bovine kidney cells [146]; antioxidant and iron chelation activities [147]; antimicrobial and antioxidant activities [148]	to support digestion [40]; hepaticoprotective [149]; digestive, stomach-ache, diarrhoea [150]	[40,69,121,123,129,145,146,147,148,149,150]
***A. tricolor* L. (syn. *A. gangeticus* L., *A. polygamus* L., *A. mangostanus* L.)**	aerial parts, crude extract of leaves, leaves, roots, roots, seeds, shoots, stems, whole plant, whole plant	antioxidant and free radical-scavenging activities [41]; antioxidant, antidiabetic, anti-inflammatory, diuretic, antimicrobial (bacterial, fungal), laxative, hepatoprotective, antimalarial, anti-ulcer, antipyretic and antinociceptive (analgesic) activities; immunomodulatory effects [113]; antioxidant and neuroprotective activities [115]; neuroprotective and anti-inflammatory activities [116]; antioxidant, anti-inflammatory, and antimicrobial activities [121]; antimicrobial activity [151,152]	astringent; used for diarrhoea, dysentery, haemorrhage, and for mouth/throat ulcers [42]; for digestion and general health [44]; used as laxative, diuretic; for fever, inflammation, diabetes, skin disorders, respiratory relief, anti-snake-venom use, and digestive issues [113]; used for fever, debility, antiseptic applications; as laxative, emollient, diuretic, spasmolytic; for allergic asthma/rhinitis (pollen), menstrual disorders (root), gastrointestinal issues, cough, bronchitis; externally emollient [122]	[41,42,44,113,115,116,121,122,151,152]
***A. spinosus* L.**	aerial parts, leaves, roots, whole plants	antioxidant, anti-inflammatory, antimicrobial, antidiabetic activities [9]; antidiabetic, anti-cholesterolemic [118]; hepatoprotective, diuretic and antidepressant activities [121]; antifungal activities [153]; antimalarial activities [154]; hepatoprotective and antioxidant activities [155]; antidepressant activity [156]; cytotoxic impact and antioxidant activity [157]; antioxidant, anticancer, antiviral and anthelmintic activities [158]	as antidote/anti-snake venom [9]; for digestive discomfort and general health [44]; used as cooling stomachic and emollient; for biliousness and haemorrhagic diathesis [122]	[9,44,118,121,122,153,154,155,156,157,158]
***A. viridis* L.**	leaves, seeds, young plants	gastroprotective, anti-ulcer, anti-inflammatory, and anticancer activities, with potential benefits for cardiovascular and degenerative diseases [14]; antioxidant activity [41]; antioxidant and anticancer activities [73]; antihyperglycemic and antioxidant activities [121]; antioxidant activity, cytotoxic impact [157]; antioxidant, anticancer, anti-inflammatory and antimicrobial activities [159]; antioxidant and anti-inflammatory activities [160]; hepatoprotective and antioxidant activities [161]	diuretic, purgative; used for inflammations, boils, abscesses, gonorrhoea, orchitis, haemorrhoids, intestinal pain, dysentery, anaemia, remugue; leaves considered febrifugal [44]	[14,41,44,73,121,157,159,160,161]

Antioxidant activity has been reported for the majority of the investigated *Amaranthus* species. A substantial proportion of the studied species also exhibit anti-inflammatory, antidiabetic, anticancer, and antimicrobial properties. In contrast, hepatoprotective and antihypertensive activities are less frequently reported. Highly bioactive compounds with diverse biological activities have been identified in *A. spinosus*, *A. viridis*, *A. tricolor*, and *A. blitum*. Some of these activities, including anthelmintic effects, have been reported particularly for *A. spinosus* [9].

The broad spectrum of activities may suggest potential synergistic effects between different groups of phytochemicals, making these species particularly promising for pharmacological studies and for the development of natural therapeutic agents with multifunctional activity. *Amaranthus hypochondriacus*, *A. hybridus*, and *A. caudatus* were shown to possess antioxidant and anti-inflammatory activities, likely associated with the presence of phenolic acids, flavonoids, and other bioactive molecules [60,110,143,162]. Some of these species also show antidiabetic and antimicrobial activity, further expanding their potential applications in medicine and the food industry. The similarities in observed bioactivities among these species may be related to their phylogenetic proximity and similar chemical composition. However, the low number of phytochemical studies conducted on some of the species, which could explain the limited data on their biological activities, is a barrier to a broader application of amaranths. Environmental factors such as climate, soil conditions, and agricultural practices may also influence the accumulation of bioactive compounds; hence, it is essential to intensify biomedical research so as to diversify the utilisation of amaranths, including upcycling agricultural waste, especially of weedy amaranths.

Different plant parts of amaranths are prescribed in traditional medicines as laxatives, diuretics, antidiabetics, antipyretics, anti-snake venom agents, antileprotic, antigonorrhoeic, expectorants, and respiratory relief in acute bronchitis, to name a few; however, verification of their efficacy requires additional research. Consumption of amaranth-based foods has been associated with various health benefits, including improved cardiovascular health, enhanced digestion, and the management of chronic diseases such as diabetes and obesity [47,163]. In several African countries, amaranth was shown to be an important dietary component in the nourishment of individuals living with HIV [93]. Furthermore, dietary supplementation with amaranth has been reported to improve aerobic performance during intense exercise, making it a suitable nutritional option during the final week of preparation for endurance events [164]. This diversity of traditional and contemporary uses highlights the potential of amaranths as a functional food with broad health-promoting properties.

### 3.3. Anti-Nutritional Compounds and Olfactory Issues

Although *Amaranthus* spp. are rich in essential nutrients, some anti-nutritional compounds may affect mineral absorption, if consumed in large quantities.

The hull of amaranth seeds contains phytates, which can bind minerals such as Ca, Zn, and Fe, thereby reducing their bioavailability and absorption [69,165]. Some studies show that amaranths (both grain and leafy forms) contain significant amounts of oxalates, especially in soluble form, which can reduce Ca bioavailability, and in susceptible individuals may contribute to a risk of calcium-oxalate kidney stones. However, much of the oxalate in grain amaranths is insoluble, and cooking or other processing (e.g., discarding cooking water) can reduce them it in the edible portions [166]. Additionally, saponins present in the seed hull impart a bitter taste to products, and can cause digestive discomfort or interfere with nutrient uptake [167]. Amaranth leaves sometimes accumulate nitrates, which, due to their toxicity at high levels, can pose risks to infants and individuals with specific health conditions [168]. Heat treatment, such as boiling followed by discarding the cooking water, has been shown to effectively reduce harmful oxalates and nitrates, improving the safety, especially of amaranths, collected from ruderal terrains [169,170]. Still, Martinez-Lopez et al. [62] noted that anti-nutritional compounds occur at relatively low levels and generally do not cause adverse effects under normal consumption patterns.

The sensory profile of amaranths (attributes such as aroma, flavour, texture, aftertaste) has been shown in multiple studies to significantly affect consumers’ acceptance [171]. Volatile aromatic compounds identified in amaranth seeds and leaves include aldehydes, ketones, and esters, which contribute to its nutty, earthy, grassy, and floral flavours. However, certain varieties of amaranth may exhibit a bitter taste or strong odour that may not appeal to certain consumers. Olfactory issues are discussed by several authors, proposing various processing techniques, such as soaking, fermentation, or heat treatment, in order to minimise these negative attributes and achieve a more pleasant taste and aroma [93,172]. For example, canned amaranth leaves are described as sweet with popcorn-like aroma and are preferred by young consumers over more bitter or astringent leafy vegetables [60]. In bakery products, blends with moderate levels of amaranth flour maintain acceptability up to certain substitution levels, beyond which the changes in texture and taste may reduce consumers’ preference [53,83,85].

## 4. Perspectives for Adaptation to Climate Change

Severe droughts, heat waves, floods, and soil degradation are only a part of the climate change extremes. Amaranth have gained recognition that it can thrive under adverse environmental conditions, thus ensuring stable yields, which is crucial for a consistent supply of its nutritionally rich seeds and leaves [173,174]. *Amaranthus* species, as C4 plants, possess remarkable drought-resistance mechanisms, such as osmotic adjustment and high root-to-shoot ratios, allowing them to maintain turgor pressure and sustain dry matter production during periods of water scarcity [175]. Therefore, Chivenge et al. [173] emphasised the potential of amaranths as crops for semi-arid regions, thus contributing to food security in such environments. Jamalluddin et al. [176] further identified diverse drought-tolerance traits in *A. tricolor* germplasm through physiological screenings, demonstrating its suitability for cultivation in regions experiencing climate variability. Beyond drought tolerance, amaranths also adapt well to poor soil conditions, i.e., they can thrive in low-fertility soils, thus becoming a viable option for areas with compromised soil quality [177,178].

Amaranths are not only promising climate-resilient crops, but exposure to abiotic stresses such as drought can enhance the accumulation of bioactive compounds in the plants, including phenolic antioxidants and flavonoids, thus increasing the antioxidant, anti-inflammatory, and cardiovascular-protective effects of both amaranth seeds and leaves [179]. Molecular breeding and genomics can steer the intraspecific diversity of *Amaranthus* towards the development of improved cultivars that combine enhanced stress tolerance with superior nutritional quality, such as increased protein content and micronutrient density, minerals, vitamin C, phenolics, and flavonoids [179,180,181]. For example, Shukla et al. [182] identified *A. tricolor* as a promising species for incorporation into hybridisation programmes aimed at producing high-yielding leaf varieties with superior nutritional profiles.

Some amaranths not only contribute to ecosystem resilience but also enhance soil health and biodiversity in agroforestry, intercropping, and organic farming (e.g., *A. cruentus*) [183]. These practices can improve nutrient cycling and reduce reliance on chemical inputs, thereby promoting the production of organic, high-quality amaranth biomass with elevated levels of proteins, vitamins, and antioxidants.

Thanks to their dual role as a climate-resilient and nutritionally valuable crop amaranths have become important candidates for future resilient and sustainable food crops. Their promotion can support diversified diets rich in plant-based proteins and functional compounds, while contributing to agricultural sustainability and climate change mitigation.

## 5. Relevance of the Topic and Current Research

Amaranths has attracted growing global attention due to its exceptional nutritional value, functional properties, ecological resilience and agricultural versatility. Traditionally consumed as leafy vegetables or pseudo-cereals, these species are now being reconsidered in the context of modern functional foods and as an opportunity for sustainable agriculture that responds to the need for adaptation to climate change and combating malnutrition and food insecurity in areas severely impacted by adverse climate conditions.

Nowadays, amaranths are increasingly considered as a source for the production of foods and beverages with health and wellness benefits. However, certain communities continue to regard it primarily as a traditional or cultural ingredient [121,184]. Diving deeper into the traditional ecological knowledge preserved by these communities can disclose ways of cultivation, preparation and consumption of amaranths that can further boost its modern utilisation. Thus, continued development and nutritional evaluation of amaranth-containing products may enhance its image as a functional food with deep cultural roots and modern relevance. The need to reduce the negative impact of agriculture on the climate accelerates the ongoing process of protein transition, i.e., replacement of animal protein with plant-based alternatives [185]. Amaranth, with its diverse profile not only of proteins but also of other valuable macro- and micronutrients, have increasingly become sources of nutriceuticals that quickly find their place in the global food chain [180,186].

Combatting hunger and malnutrition, especially in areas severely impacted by climate change, political turmoil and social injustice, is among the priorities of world organisations like the Food and Agriculture Organisation (FAO), the United Nations (UN) and the World Health Organisation (WHO). FAO have promoted amaranth as a crop with high potential to become a part of modern sustainable food strategies based on the cultural re-engagement with ancient food systems and historical usage of amaranths [187].

A comprehensive understanding of the nutritional, anti-nutritional, and sensory characteristics of amaranths is essential to maximise their potential as sustainable food resources. Reducing anti-nutritional factors such as phytates, oxalates, and nitrates through optimised cultivation and processing techniques, together with preserving or enhancing nutrient bioavailability, contributes to the diversification of food choices that boost human health. The enhancement of the sensory attributes of amaranth-based products also plays a vital role in increasing consumers’ acceptance and their market integration. In so doing, the agriculture and food industries align with the principles of sustainable food production and the United Nations Sustainable Development Goals (SDGs), particularly SDG 2 (Zero Hunger), SDG 3 (Good Health and Well-Being), SDG 12 (Responsible Consumption and Production), and SDG 13 (Climate Action) [188].

## 6. Future Directions and Conclusions

Research on Amaranthus is particularly important for addressing pressing challenges such as malnutrition, food insecurity, and climate-related agricultural stress. Bridging traditional food culture with modern nutrition science brings forward the importance of traditional knowledge as a source for food innovations demanded by challenging environmental and socio-economic conditions. The diverse modes of traditional consumption of amaranths enable the development of innovative food products that combine nutritional value with culinary appeal. This combination of multifunctionality and adaptability positions amaranths as highly promising crops for sustainable food systems and health-oriented diets. Addressing these challenges will enhance the nutritional and functional qualities of amaranth-derived products and strengthen their contribution to sustainable food systems. Future studies should focus on genetic improvement, post-harvest management, and product innovation to promote wider adoption of this resilient crop.

Being rich in high-quality proteins, essential amino acids, dietary fibre, vitamins, and minerals, amaranths are promising candidates for boosting human health and nutrition. Their bioactive compounds, including flavonoids, phenolic acids, and peptides, exhibit antioxidant, anti-inflammatory, and hypocholesterolemic effects, underscoring their potential role in preventing non-communicable diseases. Anti-nutritional factors, such as saponins, phytates, and oxalates, are also being investigated to assess their influence on nutrient bioavailability and overall health outcomes. Further studies on these topics, together with improving the organoleptic characteristics of amaranth-containing foods, would enhance our knowledge on the nutritional capacity of amaranths and reveal new pathways to amaranth-based functional food products.

Agronomic research emphasises the genus’s adaptability to diverse climatic conditions, establishing it as a crop resilient under environmental stress. Integrating traditional and contemporary agricultural knowledge would support sustainable, climate-resilient agriculture that can contribute to the alleviation of food insecurity and malnutrition, especially in arid zones more prone to climate adversities. Strategic investment in molecular breeding, personalised nutrition frameworks, and circular bioeconomy pathways would catalyse amaranth’s transition from an underutilised to an innovative crop that bridges climate resilience, nutrition, and sustainability.

Our review, together with similar works, shows that interdisciplinary collaboration in plant science, food systems innovation, and health and well-being will fully disclose the potential of *Amaranthus* spp. as a crop of the future.

## Figures and Tables

**Figure 1 foods-15-00130-f001:**
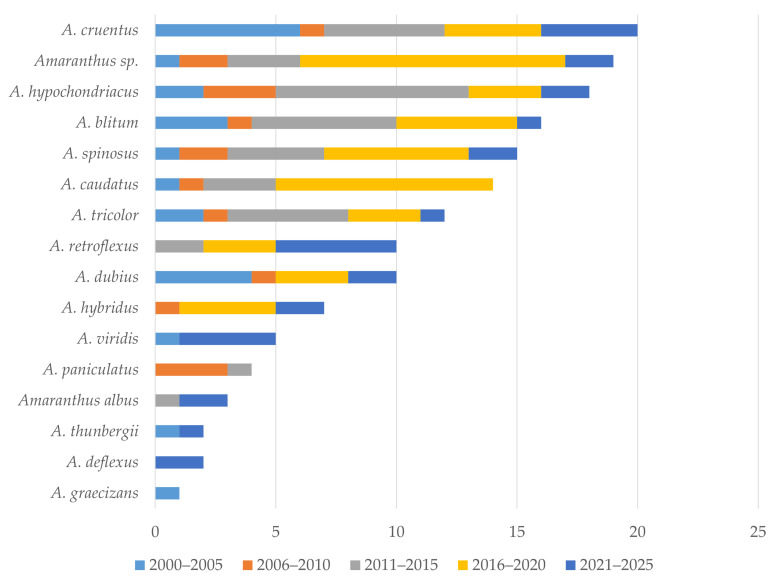
Chronological distribution of publications for different *Amaranthus* species from 2000 to 2025.

**Figure 2 foods-15-00130-f002:**
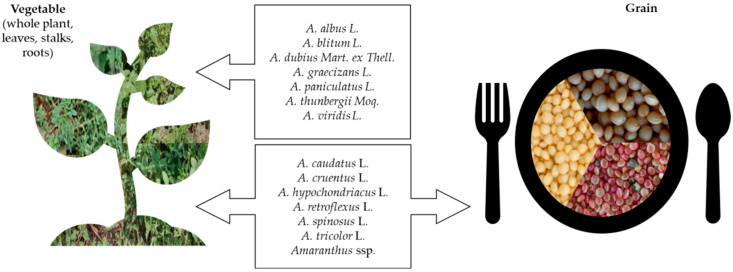
Predominant usage of *Amaranthus* species as food.

**Figure 3 foods-15-00130-f003:**
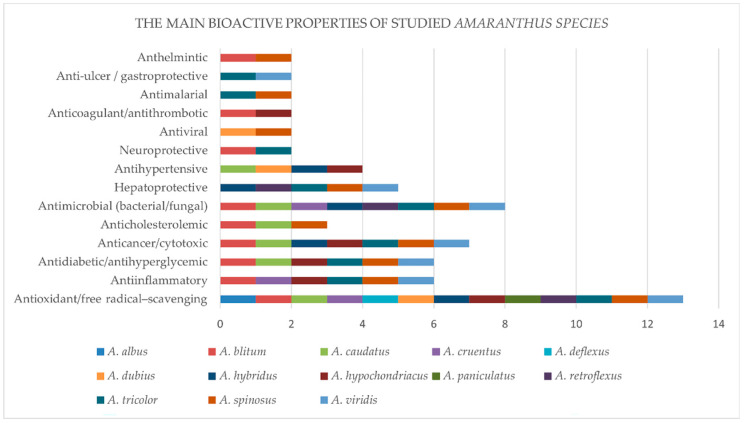
The main biological activities of the investigated *Amaranthus* species.

## Data Availability

No new data were created or analysed in this study. Data sharing is not applicable to this article.

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
