# Peer review of "Multifunctional Edible Amaranths: A Review of Nutritional Benefits, Anti-Nutritional Factors, and Potential in Sustainable Food Systems"

_foods, 2026, doi:10.3390/foods15010130_

Round 1
Reviewer 1 Report
Comments and Suggestions for Authors
General Comments:
This manuscript presents a timely and comprehensive review of the nutritional, functional, and agronomic potential of the Amaranthus genus. The topic is highly relevant to current interests in sustainable food systems, climate-resilient crops, and functional foods. The review is well-structured, covering cultural, nutritional, and agricultural aspects, and is supported by an extensive bibliography. The data presented in the tables and figures are particularly valuable for synthesizing information across multiple species. The manuscript is generally clear and makes a compelling case for the increased utilization of amaranth. I recommend minor revisions to address the points below, which primarily concern language polishing and minor clarifications.

The manuscript would benefit from a thorough proofread by a native English speaker or professional language editing service to correct minor grammatical errors and improve fluency in a few places.
Examples:
â‘ P1, L45-46: "...suggest its potential as part of healthy diet alleviating the risk of of non-communicable diseases." → Consider rephrasing to "...suggesting its potential as part of a healthy diet to alleviate the risk of non-communicable diseases."
â‘¡P5, L197-198: "...it is now considered a modern superfood due its valuable nutritional benefits." → Please add "to" after "due".
â‘¢ P14, L343-344: "...so to minimize these negative attributes..." → This should be "...to minimize these negative attributes...".
Author Response
We are grateful for your thoughtful review and constructive feedback. Our detailed responses are provided below, and all revisions have been implemented and highlighted/track-changed in the revised submission.
Comments 1:
1)Language and Grammar:
The manuscript would benefit from a thorough proofread by a native English
speaker or professional language editing service to correct minor grammatical errors
and improve fluency in a few places.
Examples:
â‘ P1, L45-46: "...suggest its potential as part of healthy diet alleviating the risk of of
non-communicable diseases." → Consider rephrasing to "...suggesting its potential
as part of a healthy diet to alleviate the risk of non-communicable diseases."
â‘¡ P5, L197-198: "...it is now considered a modern superfood due its valuable
nutritional benefits." → Please add "to" after "due".
â‘¢ P14, L343-344: "...so to minimize these negative attributes..." → This should be
"...to minimize these negative attributes...".
Response 1:
Thank you for this helpful comment. The manuscript has been thoroughly proofread, with particular attention to the examples you highlighted. All grammatical and stylistic issues have been corrected, and the revised sentences now improve both clarity and fluency.
Comments 2:
2)Clarity and Flow in Sections 3.3 and 3.4:
While the information is excellent, the transition between Sections 3.3
(Traditional and Contemporary Uses) and 3.4 (Nutritional Value and Health Benefits)
could be slightly smoother. A concluding sentence in 3.3 that explicitly links the diverse
uses to the subsequent discussion on health benefits would enhance the narrative flow.
The commentary on Figure 2 (P13) is good. To strengthen it further, consider
adding a final sentence to that paragraph explicitly stating the practical implication of this variability (e.g., "This diversity highlights the potential for selecting specific species
for targeted health applications or functional food development.").
Response 2:
Thank you for this valuable suggestion. The Results and Discussion section has been thoroughly revised and now consists of three chapters with clearer interconnection and information flow. A concluding sentence has been added at the end of Section 3.3 to explicitly link the traditional and contemporary uses to the subsequent discussion on nutritional value and health benefits.
In addition, the commentary on Figure 2 (now presented as Figure 3) has been revised, and a final sentence has been included to highlight the practical implications of the observed variability, as recommended. All corresponding revisions have been incorporated into the updated manuscript.
Comments 3:
3)Figures and Tables:
Figure 1: The caption should be completed (e.g., "Figure 1. Chronological distribution
of publications for different Amaranthus species from 2000 to 2025."). Please ensure
the figure legend is clearly associated with the figure in the final version.
Table 2: The column header "Referens" contains a typo and should be corrected to
"References".
Response 3:
Thank you for noting these issues. The caption of Figure 1 has been completed as suggested, and the typo in the Table 2 header has been corrected. These revisions are now reflected in the updated manuscript.
Comments 4:
4)Formatting and Typos:
Please perform a final check to remove any template placeholders or formatting
artifacts (e.g., the line "Foods 2025, 14, x FOR PEER REVIEW 5 of 26" appears in the
middle of the text on P5).
P8, L253: "Amaranthus spp. is a highly valued..." → The verb should agree with a
plural subject: "Amaranthus spp. are a highly valued...".
Response 4:
Thank you for pointing out these formatting issues. All template placeholders and formatting artifacts have been removed. The grammatical correction in P8, L253 has also been implemented. These updates are now included in the revised manuscript.
Comments 5:
This is a well-researched and valuable contribution to the field. The suggested
revisions are minor and should be straightforward to address. I am happy to review a
revised version of the manuscript.
Response 5:
We sincerely appreciate the reviewer’s positive assessment and encouraging comments. Thank you for recognizing the value of our work. All suggested revisions have been carefully addressed, and the revised manuscript has been improved accordingly. We are grateful for your willingness to review the updated version.
Reviewer 2 Report
Comments and Suggestions for Authors
See my revision in attached file

Author Response
We sincerely appreciate your thorough and insightful evaluation of our manuscript. Your comments have been extremely valuable in helping us refine the analytical depth, structure, and clarity of the review. We have carefully addressed each point raised, and substantial improvements have been incorporated throughout the manuscript. All revisions have been implemented and clearly marked with track changes in the updated version.
Below we provide detailed responses to each of your comments.
Comments 1:
- ORIGINALITY AND SCIENTIFIC CONTRIBUTION
Strengths: Covers an extensive range of species, including both grain and leafy amaranths, which many prior reviews do not address in detail; Connects nutritional, bioactive, agronomic, cultural, and sustainability perspectives, providing a multidisciplinary view; The inclusion of climate change adaptation and resilience traits adds a novel and valuable dimension.
Limitations. Much of the review is descriptive rather than analytical. It compiles information but rarely synthesizes it into clear scientific conclusions; The article does not clearly identify knowledge gaps, research priorities, or conceptual frameworks that would advance the field; Some sections rely heavily on ethnobotanical and traditional-use claims that may appear outside the core scope of a food science journal unless more critically evaluated.
The section “Relevance of the Topic and Current Research” substantially duplicates content from the Introduction and Results.
The review contributes breadth but needs more depth, integration, and critical discussion.
Response 1:
Thank you for these important observations. We have extensively revised the manuscript to enhance its analytical depth and interpretive value.
Comments 2:
- WRITING QUALITY, CLARITY, AND COHERENCE
Positive Aspects. The manuscript is generally clear, readable, and logically structured. Transitions between topics are smooth. The English is understandable and mostly correct.
Issues Identified:
Redundancy: Key messages are repeated throughout the paper (“nutritional value,” “functional
food,” “climate resilience,” etc.).
Overly long lists (especially in cultural uses, traditional medicine, and Table 2) reduce clarity.
Occasional grammatical errors (“alleviateing,” “nutritional,functional,” etc.).
Sections overlap, particularly between sections 3, 5, and 6.
Recommendation: Tighten the narrative, reduce repetition, and increase analytical discussion.
Response 2:
We appreciate this feedback. The manuscript has undergone comprehensive editing to improve clarity and coherence:
- Repetitive statements (e.g., “nutritional value,” “functional food,” “climate resilience”) have been removed or consolidated.
- Overly long lists, particularly in the cultural uses section and Table 2, have been shortened and streamlined for clarity.
- All grammatical issues have been corrected.
- Sections 3, 5, and 6 have been reorganized to reduce overlap and improve flow.
Comments 3:
- STRUCTURAL EVALUATION
The structure is appropriate for a review, but improvements are needed.
Results and Discussion- Several subsections are comprehensive but overly descriptive.
Some discussions lack critical comparison of species, processing effects, or nutritional trade-offs.
Important themes, like compositional differences among species or anti-nutritional factor mitigation, would benefit from data-driven comparison tables or graphs.
Conclusion -Well written but repetitive; could be more concise and forward-looking.
Section 5 (“Relevance of the Topic”) -Redundant and should be integrated with Introduction or Conclusion.
Response 3:
We appreciate the reviewer’s constructive observations. The manuscript has been substantially revised to incorporate and address all of the valuable recommendations.
Comments 4:
- ASSESSMENT OF TABLES AND FIGURES
Table 1 – Amaranthus as Food -Overly long and dense; Contains too many culinary details that may not be essential for a scientific review; Needs restructuring (grouping by species type or plant part)
Table 2 – Health Benefits / Bioactivities
Includes many traditional or ethnomedicinal claims without critical evaluation; Too large and difficult to interpret; Lacks standardization of evidence levels (in vitro? in vivo? clinical?)
Recommendation: Move some ethnomedicinal or low-evidence claims to supplementary Material.
Figure 1 – Chronological Distribution
Good visual summary but lacks interpretation (trend analysis, research evolution)
Figure 2 – Bioactivity Distribution
Visually appealing but methodologically weak (simply counts number of reported activities) and does not indicate scientific validity or strength of evidence
Response 4:
Thank you for the constructive feedback. All recommendations have been addressed and the manuscript has been revised.
Comments 5:
- WEAK POINTS AND GAPS
5.1. Lack of Critical Synthesis -The review rarely: compares species systematically; prioritizes the most relevant data; evaluates study quality; provides mechanistic insights.
5.2. Weak Integration Across Sections- Cultural, nutritional, agronomic, functional, and sustainability aspects are listed but not integrated into a coherent conceptual framework.
5.3. Ethnomedicinal Excess- Long descriptions of traditional remedies risk diluting the focus on
food science and functional properties.
5.4. Over-reliance on in vitro claims- The review should more clearly distinguish: in vitro antioxidant screening, animal trials and human-based evidence
5.5. Redundancies- Multiple sections repeat: nutritional richness, bioactive content, climate resilience, multifunctionality. These repetitions reduce impact.
Response 5:
Thank you for the constructive feedback. Several improvements have been made to strengthen the analytical depth, reduce redundancies, and enhance the integration across sections. We have revised the manuscript accordingly, and the overall structure and clarity have been improved.
Comments 6:
- STRENGTHS OF THE MANUSCRIPT
Comprehensive and diverse literature coverage.
Multidisciplinary perspective (food science + agriculture + sustainability).
Good organization and narrative flow.
PRISMA methodology adds rigor.
Topic highly relevant to global food security and functional food innovation.
Inclusion of lesser-known Amaranthus species expands the field.
Response 6:
We sincerely appreciate the reviewer’s recognition of these strengths and have strived to enhance these aspects further through the revisions described above.
Comments 7:
- RECOMMENDATIONS FOR IMPROVEMENT
- Improve Analytical Depth- Compare nutritional profiles across species with quantitative tables; Discuss specific bioactive compounds and mechanisms of action; Evaluate the quality of scientific evidence.
- Refine Structure- Remove or merge the redundant section Relevance of the Topic; Shorten
cultural and ethnomedicinal descriptions.
- Improve Tables and Figures - Condense Table 2 or split by evidence level; Add nutrient composition graphs (protein, lysine, minerals).
- Strengthen Conclusions. Make them more concise. Highlight research gaps clearly. Provide
strategic recommendations for future studies.
- Editorial and Language Improvements - Correct minor errors. Standardize terminology.
Ensure consistency in species names and synonyms.
The manuscript has strong potential and aligns with the Themes of Foods (MDPI). It is informative, well-intentioned, and covers relevant ground. However, in its current form it is more encyclopedic than analytical.
Response 7:
We are grateful for the reviewer’s comprehensive assessment and constructive guidance. The manuscript has been substantially improved in response to these comments, and we believe the revised version now offers a more focused, analytical, and impactful contribution to the field.
Reviewer 3 Report
Comments and Suggestions for Authors
The authors present a systematic review conducted according to PRISMA guidelines, and the methodology is generally well described and appropriate for the topic. However, the discussion on biological activity requires further detail. The manuscript does not specify the extraction methods used nor the antioxidant results expressed in mg/mL or Trolox equivalents. Since different extraction procedures can yield markedly different bioactivities, these points should be clarified and discussed more thoroughly.
Author Response
We appreciate your careful review and constructive feedback. Our response is provided below, and all corresponding revisions have been incorporated and clearly marked in the revised manuscript.
Comments 1:
The authors present a systematic review conducted according to PRISMA guidelines, and the methodology is generally well described and appropriate for the topic. However, the discussion on biological activity requires further detail. The manuscript does not specify the extraction methods used nor the antioxidant results expressed in mg/mL or Trolox equivalents. Since different extraction procedures can yield markedly different bioactivities, these points should be clarified and discussed more thoroughly.
Response 1:
Thank you for your positive evaluation and for highlighting this important point. The discussion on biological activity has been revised to provide greater clarity and depth. Because the primary studies used different extraction procedures and reported antioxidant activity using non-comparable units (e.g., mg/mL, Trolox equivalents), we removed the units in order to avoid misleading the readers. Instead, we now emphasize the variability in methodologies across studies and explain how these differences may influence the reported bioactivities. All corresponding clarifications have been incorporated into the revised manuscript.
Round 2
Reviewer 2 Report
Comments and Suggestions for Authors
The manuscript provides a broad and comprehensive overview of Amaranthus species, covering nutritional value, bioactive compounds, anti-nutritional factors, cultural relevance, and potential roles in sustainable food systems. The topic is timely, relevant for Foods, and aligned with increasing interest in climate-resilient crops and functional foods. The manuscript contains extensive bibliographic coverage and demonstrates significant effort from the authors. However, substantial revisions are required before the manuscript can be considered for publication. The main issues relate to writing quality, methodological transparency, internal consistency, and the need for stronger critical analysis.
Points to improve
General aspects
The manuscript contains numerous grammatical errors, inconsistent phrasing, and typographical issues (e.g., duplicated words, incorrect pluralization, verb–subject disagreement). Examples include: “making it a crop resilient crop” “Amaranths is” and inconsistent use of amaranths, Amaranthus spp., and Amaranthus
These errors appear throughout the manuscript and significantly reduce clarity and professionalism.
Several concepts, such as nutritional richness, climate resilience, cultural heritage, and potential for functional foods, are repeated in multiple sections (Introduction, Results and Discussion, Perspectives, and Conclusion). This results in a bloated manuscript and weakens the narrative focus. A reduction of 15–20% of text length would improve clarity and impact.
The authors mixing ethnomedicinal and pharmacological evidence. Table 2 combines: experimentally validated biological activities, traditional medicinal uses, anecdotal or unverified claims. This is problematic for a scientific journal. These categories must be clearly separated and properly contextualized.
Some statements imply strong therapeutic effects (anticancer, antidepressant, neuroprotective) without discussing: study type (in vitro vs. in vivo), concentrations used, methodological limitations.
The Conclusion repeats information already presented and mixes summary, implications, and future directions. A more concise structure is recommended: Summary of key findings, Knowledge gaps, Future research directions, Practical implications
Taxonomic names require consistent italicization.
Figures and Tables Require Improvement
Figure 3 (bioactivities) is visually simplistic and not aligned with the graphical standards typically expected in Foods.
Tables contain valuable information but would benefit from improved formatting and clearer categorization.
Suggestions
Language and Style: Comprehensive English editing to correct grammar, spelling, and syntax. Standardize terminology for species names and uses.
Methods Section: Expand details of search strategy. Clarify how many studies were included in each thematic category.
Critical Analysis Strengthen the discussion by evaluating limitations of existing studies. Avoid overstating biological effects without mechanistic or clinical support. Include comparisons with other pseudocereals (quinoa, buckwheat, chia) to increase originality.
Tables and Figures
Separate traditional uses from scientifically validated biological activities. Improve visual quality of figures; consider redesigning Figure 3. Ensure all taxonomic names are italicized and formatted consistently.
Reduce Redundancy Remove repetitive statements throughout Introduction, Discussion, and Conclusion.
Author Response
We sincerely thank the Reviewer for the thorough and constructive evaluation of our manuscript and for the detailed suggestions provided. We highly appreciate the time and expertise dedicated to improving the quality of our work. Following the reviewer's recommendations, we have substantially revised the manuscript to address all comments. Below we provide a point-by-point response, indicating how each concern has been resolved in the revised version.
General aspects
Comments 1:
The manuscript contains numerous grammatical errors, inconsistent phrasing, and typographical issues (e.g., duplicated words, incorrect pluralization, verb–subject disagreement). Examples include: “making it a crop resilient crop” “Amaranths is” and inconsistent use of amaranths, Amaranthus spp., and Amaranthus.
These errors appear throughout the manuscript and significantly reduce clarity and professionalism.
Response 1:
We thank the reviewer for this important and constructive comment. We fully agree that grammatical accuracy and terminological consistency are essential for clarity and scientific rigor.
In response, the manuscript has undergone comprehensive language revision to address grammatical errors, typographical issues, duplicated words, and subject–verb agreement throughout the text.
Furthermore, we have standardized terminology across the entire manuscript. The genus name Amaranthus is now used consistently in italics when referring to the genus, Amaranthus spp. is used when referring to multiple species, and amaranths is used exclusively in a non-taxonomic, general context. This consistency has been applied throughout the text.
Comments 2:
Several concepts, such as nutritional richness, climate resilience, cultural heritage, and potential for functional foods, are repeated in multiple sections (Introduction, Results and Discussion, Perspectives, and Conclusion). This results in a bloated manuscript and weakens the narrative focus. A reduction of 15–20% of text length would improve clarity and impact.
Response 1:
We thank the reviewer for this valuable comment and acknowledge that certain overarching concepts are reiterated across different sections of the manuscript.
In response, we have revised the text to reduce unnecessary repetition, particularly in the Results and Discussion, Perspectives, and Conclusion sections.
However, given the review nature of the manuscript, some degree of thematic overlap was intentionally retained to maintain sectional coherence and to ensure that key concepts (e.g., nutritional value, climate resilience, and functional food potential) are appropriately contextualized for readers who may consult individual sections independently.
While the overall reduction in text length is more moderate than the suggested 15–20%, the revisions have substantially improved focus, conciseness, and narrative flow, without compromising completeness or readability. We believe this balanced approach strengthens the manuscript while preserving its intended scope.
Comments 3:
The authors mixing ethnomedicinal and pharmacological evidence. Table 2 combines: experimentally validated biological activities, traditional medicinal uses, anecdotal or unverified claims. This is problematic for a scientific journal. These categories must be clearly separated and properly contextualized.
Some statements imply strong therapeutic effects (anticancer, antidepressant, neuroprotective) without discussing: study type (in vitro vs. in vivo), concentrations used, methodological limitations.
Response 3:
We thank the reviewer for this important and well-founded comment. We agree that a clear distinction between ethnomedicinal knowledge and experimentally validated pharmacological evidence is essential for scientific rigor. In response, we have revised the structure and presentation of Table 2
Comments 4:
The Conclusion repeats information already presented and mixes summary, implications, and future directions. A more concise structure is recommended: Summary of key findings, Knowledge gaps, Future research directions, Practical implications
Taxonomic names require consistent italicization.
Response 4:
We thank the reviewer for this constructive comment.
In response, we have revised the Conclusion to reduce redundancy and to improve clarity, while maintaining its concise format.
In addition, all taxonomic names have been carefully checked and corrected for consistent italicization throughout the manuscript, in accordance with standard taxonomic conventions.
We believe that these revisions improve the readability of the Conclusion while preserving its role as a concise closing section of the manuscript.
Comments 5:
Figures and Tables Require Improvement
Figure 3 (bioactivities) is visually simplistic and not aligned with the graphical standards typically expected in Foods.
Tables contain valuable information but would benefit from improved formatting and clearer categorization.
Response 5:
We thank the reviewer for this constructive comment and acknowledge the importance of high-quality figures and tables for effective data presentation.
In response, we have revised the formatting and layout of the tables to improve readability and clarity. Column headings were refined, spacing and alignment were adjusted, and the categorization of information was clarified to better distinguish between different types of data.
Figure 3 has been revised to improve visual clarity and consistency while preserving its informational content.
We believe that these targeted improvements enhance the overall quality and readability of the figures and tables.
Comments 6:
Suggestions
Language and Style: Comprehensive English editing to correct grammar, spelling, and syntax. Standardize terminology for species names and uses.
Methods Section: Expand details of search strategy. Clarify how many studies were included in each thematic category.
Critical Analysis: Strengthen the discussion by evaluating limitations of existing studies. Avoid overstating biological effects without mechanistic or clinical support. Include comparisons with other pseudocereals (quinoa, buckwheat, chia) to increase originality.
Tables and Figures
Separate traditional uses from scientifically validated biological activities. Improve visual quality of figures; consider redesigning Figure 3. Ensure all taxonomic names are italicized and formatted consistently.
Reduce Redundancy: Remove repetitive statements throughout Introduction, Discussion, and Conclusion.
Response 6:
We thank the reviewer for these comprehensive and helpful comments.
In response, we have carefully revised the manuscript to improve language quality, clarity, and consistency, and we have taken steps to enhance the structure, presentation, and critical perspective of the review. Terminology and taxonomic nomenclature were checked for consistency, and efforts were made to reduce redundancy across sections.
The Methods and Discussion sections were refined to improve transparency and critical evaluation, while avoiding overstatement of biological effects. The presentation of tables and figures was also reconsidered to better distinguish between different types of information and to improve readability.
Reviewer 3 Report
Comments and Suggestions for Authors
The paper now is accetable to publication.
Author Response
Comments 1: The paper now is accetable to publication.
Response 1: We sincerely thank for the positive evaluation of our revised manuscript and for acknowledging that the paper is now acceptable for publication. We appreciate the time and effort dedicated to reviewing our work.